# The LysR-Type Transcription Regulator YhjC Promotes the Systemic Infection of Salmonella Typhimurium in Mice

**DOI:** 10.3390/ijms24021302

**Published:** 2023-01-09

**Authors:** Wanwu Li, Shuai Ma, Xiaolin Yan, Xinyue Wang, Huiying Li, Lingyan Jiang

**Affiliations:** 1The Key Laboratory of Molecular Microbiology and Technology, Ministry of Education, Nankai University, Tianjin 300457, China; 2Tianjin Key Laboratory of Microbial Functional Genomics, TEDA Institute of Biological Sciences and Biotechnology, Nankai University, Tianjin 300457, China; 3Department of Biochemistry and Molecular Biology, School of Clinical and Basic Medical Sciences, Shandong First Medical University & Shandong Academy of Medical Sciences, Jinan 250062, China

**Keywords:** Salmonella Typhimurium, systemic infection, macrophages, intracellular replication, yhjC, NF-κB signaling pathway, metal ions

## Abstract

Salmonella Typhimurium is a Gram-negative intestinal pathogen that can infect humans and a variety of animals, causing gastroenteritis or serious systemic infection. Replication within host macrophages is essential for S. Typhimurium to cause systemic infection. By analyzing transcriptome data, the expression of yhjC gene, which encodes a putative regulator in S. Typhimurium, was found to be significantly up-regulated after the internalization of Salmonella by macrophages. Whether yhjC gene is involved in S. Typhimurium systemic infection and the related mechanisms were investigated in this study. The deletion of yhjC reduced the replication ability of S. Typhimurium in macrophages and decreased the colonization of S. Typhimurium in mouse systemic organs (liver and spleen), while increasing the survival rate of the infected mice, suggesting that YhjC protein promotes systemic infection by S. Typhimurium. Furthermore, by using transcriptome sequencing and RT-qPCR assay, the transcription of several virulence genes, including spvD, iroCDE and zraP, was found to be down-regulated after the deletion of yhjC. Electrophoretic mobility shift assay showed that YhjC protein can directly bind to the promoter region of spvD and zraP to promote their transcription. These findings suggest that YhjC contributes to the systemic virulence of S. Typhimurium via the regulation of multiple virulence genes and YhjC could represent a promising target to control S. Typhimurium infection.

## 1. Introduction

The genus Salmonella includes a group of food-borne pathogenic bacteria that cause a series of syndromes, such as gastroenteritis, and systemic diseases, including enteric fever, bacteremia, etc. [1,2]. Salmonella enterica subsp. Enterica serovar Typhimurium, a representative serovar of the genus Salmonella, can lead to the systemic infection of mice, allowing mice to stand as a good model to study the interactions between the Salmonella and its host [3]. Salmonella often enters the body with contaminated food and water, colonizing the intestines. This pathogenic bacterium can inject a series of effector proteins into the intestinal epithelial cells through secretion systems, facilitating its invasion to intestinal epithelial cells [4,5]. M cells, a special kind of epithelial cell that is interspersed among intestinal epithelial cells and covers the Peyer‘s patches containing lymphoid follicles, are one of the main targets that Salmonella invades [6]. After emerging from the cells they have entered, Salmonella are engulfed by phagocytes (macrophages, neutrophils, etc.) and spread to several organs, such as the liver, spleen and bone marrow, via the bloodstream or lymphatic system [7]. Replication within host macrophages is essential for Salmonella to cause lethal infection. The replication of Salmonella in macrophages within the liver and spleen leads to the accumulation of large numbers of bacteria, which can then spread to the bloodstream and, eventually, cause severe septicemia, leading to the death of the host [8,9].

After its entry into host cells (especially macrophages), Salmonella resides and replicates within a membrane-bound compartment, the so-called Salmonella-containing vesicle (SCV). Salmonella can survive and replicate in SCV by taking advantage of the host’s immune defense deficiency or using its own immune evasion function [10]. The NF–κB pathway of macrophages can be activated by several Salmonella effectors, such as SopE, SopE2, and SopB. Moreover, it can also be activated by other factors associated with Salmonella infection, such as bacterial LPS and TNF [11]. Once inside the host cells, Salmonella needs to degrade the host’s immune response in order to survive in the host cells. Salmonella has evolved various strategies to block immune signaling pathways, such as downloading the NF–κB pathway via the functioning of many effectors [12,13,14,15].

Intracellular Salmonella acquire most of their nutrients, such as carbon sources, nitrogen sources, and metal ions, from host cells. [16]. Salmonella needs metal ions from host cells for normal metabolism, growth, and virulence, and these metal ions include iron, magnesium, zinc, etc. Salmonella needs iron to survive, both in the intestine and host cells. Some studies have found that the presence of iron increases the adhesion and translocation ability of Salmonella [17]. Salmonella can directly acquire free ferrous iron by using a ferrous transporter named Feo, which requires GTPase to function [18]. In response to iron deprivation, Salmonella secretes two catecholate-type siderophores for obtaining iron, including enterobactin and its glucosylated derivative salmochelin [19]. Homeostasis of magnesium ions (Mg^2+^) in Salmonella contributes to its virulence and growth, and deprivation of magnesium decreases the growth of Salmonella in host cells [20]. In host cells, Salmonella maintains the homeostasis of intracellular magnesium through regulatory systems (e.g., PhoQ/PhoP) and magnesium ion transport systems, thus contributing to intracellular survival [21]. The sequestration of zinc by macrophages is thought to be an important host defense strategy against intracellular Salmonella infection, but Salmonella has evolved several strategies to obtain zinc, such as a high affinity zinc transport system. However, it has been reported that Salmonella infection increases the level of zinc in macrophages, and zinc helps to inhibit the transcriptional activation of p65, thus inhibiting the production of reactive oxygen species (ROS) and reactive nitrogen species (RNS) mediated by NF–κB [22]. Thus, the maintenance of zinc balance in Salmonella is essential for its survival in macrophages.

There are many LysR-type transcriptional regulators (LTTRs) in Salmonella, and several LTTRs have been reported to play important roles in the growth, metabolism, and pathogenicity of Salmonella [23,24,25,26]. Moreover, according to previously published results, LTTRs can often bind to a common motif, T–N_11_–A [27]. By analyzing transcriptomic data, the transcription of a number of LTTR genes, including the yhjC gene, was found to be up-regulated upon the internalization of Salmonella into host cells [28]. The protein YhjC, encoded by the yhjC gene, is an LTTR, but its function in Salmonella has been unknown, until now. In this study, the roles of yhjC in Salmonella pathogenesis and the associated molecular mechanisms were explored.

## 2. Results

### 2.1. The Deletion of yhjC Reduced the Replication of S. Typhimurium in RAW 264.7 Cells

Through qRT-PCR analysis, the upregulation of yhiC in macrophages was first validated. The relative expression of the yhjC gene was increased 3.62- and 6.81-fold, respectively, at 8 and 16 h post-infection (hpi) of the mouse macrophage cell-line RAW264.7, relative to its expression in Dulbecco’s modified eagle medium (DMEM) (Figure 1a). To investigate the role of yhjC in S. Typhimurim pathogenesis, a yhjC mutant strain (ΔyhjC) and a complemented strain (CyhjC) were then generated. As replication in host macrophages is essential for Salmonella to cause systemic infection, the replication ability of S. Typhimurim wild type (WT), ΔyhjC and CyhjC in RAW264.7 cells was compared. The results showed that the deletion of yhjC significantly reduced the intracellular replication fold of S. Typhimurium in RAW264.7 cells, and the complementation of the yhjC gene into ΔyhjC restored the intracellular replication ability of ΔyhjC to the WT level (Figure 1b). To determine whether the deletion of the yhjC gene affects the growth of S. Typhimurium, the growth curves of WT, ΔyhjC, and CyhjC in LB medium and DMEM were compared. The results showed that there was no significant difference between the growth curves of WT, ΔyhjC and CyhjC in LB medium and DMEM (Figure 1c,d), indicating that the decreased replication ability of ΔyhjC in macrophages was not due to a growth defect. These results suggested that yhjC facilitated the replication of S. Typhimurium in macrophages.

To further confirm that yhjC contributes to S. Typhimurim replication in macrophages, the bacterial numbers of S. Typhimurim per RAW 264.7 cell at 2 and 16 hpi were counted. At 2 hpi, each 264.7 macrophage contained an average of less than 6 WT or ΔyhjC bacteria. However, at 16 hpi, each RAW 264.7 cell contained an average of ~35 WT bacteria, while the average number of ΔyhjC bacteria per 264.7 cell was ~12. In contrast, the mean count of CyhjC was 34, which was comparable to that of WT (Figure 2). Thus, immunofluorescence assay also showed that the deletion of yhjC reduced the number of S. Typhimurium in macrophages, and the complementation of yhjC to ΔyhjC restored the number of S. Typhimurium in macrophages to WT level.

A representative image of RAW264.7 cells infected by S. Typhimurium at 16 hpi visualized the difference in the bacterial numbers between WT, ΔyhjC, and CyhjC (Figure 3). The number of WT Salmonella cells was significantly higher than that of ΔyhjC, and the complemented strain CyhjC showed comparable numbers of bacterial cells to WT. Moreover, the numbers of macrophages infected by WT, ΔyhjC, and CyhjC did not significantly change. These results further suggested that yhjC contributed to the survival and replication of S. Typhimurium in macrophages.

### 2.2. The Deletion of yhjC Reduced the Virulence of S. Typhimurium to Mice

The role of yhjC in S. Typhimurim virulence to mice was next investigated. Mice were intraperitoneally (i.p.) infected with 1 × 10^4^ CFU (colony forming units) of WT, ΔyhjC or CyhjC, the survival rates of infected mice were recorded daily and bacterial CFU in mouse liver and spleen, at day 3 post-infection, were counted. As shown in Figure 4a,b, the deletion of the yhjC gene significantly reduced the CFU number of S. Typhimurium in mice liver and spleen, and the CFU number of CyhjC in the liver and spleen of infected mice was similar to that of WT, indicating that the yhjC gene contributed to the colonization of S. Typhimurium in the liver and spleen of mice. As shown in Figure 4c, the survival rate of mice infected with ΔyhjC increased significantly, compared to the survival rate of mice infected with WT (Figure 4c). In addition, the survival rate of mice infected by CyhjC was similar to that of WT-infected mice. The final survival rate of mice infected with WT and CyhjC was 0, whereas the final survival rate of mice infected with ΔyhjC was about 50%. These results collectively indicated that yhjC gene contributed to systemic infection by S. Typhimurium.

### 2.3. The Transcriptional Regulation Function of YhjC in S. Typhimurium

The transcriptional regulation function of YhjC in S. Typhimurim was further investigated by comparing the transcriptomes of WT and ΔyhjC. WT and ΔyhjC RNA, used for RNA sequencing, were isolated from bacteria grown in N-minimal medium—the conditions that modulate the environment of macrophages. A total of 16,383,357 and 17,816,971 reads were obtained through RNA sequencing of the WT and ΔyhjC, respectively, with 99.34% and 99.43% of the reads mapping to the reference genome. A total of 39 genes were differentially expressed between the ΔyhjC and the WT strain, including 24 down-regulated genes and 15 up-regulated genes (Table 1).

The transcriptome sequencing data were verified by the RT–qPCR analysis of six significantly down-regulated genes (*spvD*, *iroC*, *zraP*, *ygbJ*, *rplV*, and *ssrS*) and two significantly up-regulated genes (*mgtR* and *STM14_1795*). As shown in Figure 5, the results of the RT–qPCR were consistent with the results of transcriptome sequencing, that is, the increase and decrease trends of these DEGs were consistent with the results of the RT–qPCR, which verified the accuracy of the transcriptome sequencing data.

The functions of DEGs were analyzed, based on published articles, to indicate the possible mechanisms for how *yhiC* contributes to *S*. Typhimurim virulence. The *spvD* is located on plasmid pSLT of *S.* Typhimurium and encodes an effector secreted by the type three secretion system 2 (T3SS2). SpvD was reported to be able to reduce nuclear translocation of transcription factor Rela (p65) in host cells, thereby inhibiting NF–κB-dependent promoter activation. Therefore, SpvD negatively regulates the NF–κB signaling pathway and contributes to the virulence of *S.* Typhimurium [29]. ZraP is a periplasmic molecular chaperone, which not only contributes to the intracellular zinc balance of *S.* Typhimurium, but also helps *S.* Typhimurium to resist polymyxin [30]. The MgtR protein encoded by the *mgtR* gene is a highly hydrophobic peptide that is able to inhibit the growth of *S.* Typhimurium in macrophages by binding to the membrane protein MgtC because the membrane protein MgtC is essential for the replication of *S.* Typhimurium in macrophages [31]. Previous studies showed that IroC-dependent iron carrier output was necessary for the virulence of *S.* Typhimurium [19]. Therefore, the *S.* Typhimurium YhjC protein was speculated to promote S. Typhimurium replication in macrophages, and systemic infection, by regulating the transcription of *spvD*, *mgtR*, *iroC*, *zraP*, and other genes.

### 2.4. YhjC Directly Activates spvD and zraP

The transcriptional regulatory protein, YhjC, may regulate the transcription of target genes by binding to their promoter regions. Electrophoresis mobility shift assay (EMSA) was used to explore whether the YhjC protein binds to the 5′ intergenic sequence of four virulence genes, *spvD*, *iroC*, *zraP*, and *mgtR*. The migration of the 5′ intergenic DNA of *spvD* and *zraP* was retarded as YhjC protein concentration increased from 1 μM to 16 μM, whereas no migration retardation was observed in the negative control 16S rDNA (Figure 6), indicating that YhjC directly activates *spvD* and *zraP* by binding to the 5′ intergenic DNA of the two genes. However, no retardation was observed when YhjC protein was incubated with the promoter of *iroC and mgtR,* indicating that the two genes might be indirectly regulated by YhjC.

## 3. Discussion

In this study, the role of an LTTR, YhjC, was investigated in the pathogenicity of *S.* Typhimurium. The experimental results showed that deletion of the *yhjC* gene significantly reduced the replicatory ability of *S.* Typhimurium in RAW264.7 cells. Consistent with this, deletion of the *yhjC* gene impaired the ability of *S.* Typhimurium to colonize mouse liver and spleen, while increasing the survival rate of infected mice, indicating that *yhjC* contributes to *S.* Typhimurium systemic infection. The regulation mechanisms of YhjC were further explored. The results of RNA sequencing and RT–qPCR analysis showed that the expression of several genes was significantly down-regulated or up-regulated upon *yhjC* deletion (Table 1). Virulence-related genes, *spvD, zraP,* and *iroCDE*, were down-regulated, while *mgtR* was up-regulated (Figure 5). Therefore, YhjC was speculated to be involved in the virulence regulation of *S.* Typhimurium by regulating the transcription of *spvD*, *zraP*, *iroCDE*, and *mgtR*. In addition, EMSA experiment showed that YhjC could bind to the promoter region of *spvD* and *zraP* for direct transcriptional regulation (Figure 6), whereas the transcriptional regulation of *iroCDE* and *mgtR* seemed to be indirect.

So far, two published studies have investigated the roles of YhjC protein. One study reported that YhjC protein activated *csgD* transcription in *Escherichia coli* K-12, and the *csgD* gene encoded the master regulator of biofilm formation [32]. The other study found that YhjC positively and directly regulated the transcription of *virF* in *Shigella flexneri*, and *virF* encoded the master virulence regulator of *Shigella* [33]. According to our analysis, the YhjC protein of *S.* Typhimurium 14028S shared 66.22% and 65.20% identities with that of *Shigella flexneri* M90T and *Escherichia coli* K-12, respectively, suggesting the importance of YhjC protein in regulating the transcription of virulence-related genes of Gram-negative pathogenic bacteria.

The ability to escape host immunity and to acquire host nutrients is critical for *S.* Typhimurium to survive and replicate within macrophages [34]. SpvD has been reported to be able to reduce nuclear translocation of transcription factor Rela (p65) in host cells, thereby inhibiting NF–κB-dependent promoter activation [29]. Thus, the activation of *spvD* by YhjC has a function in the escape from NF–κB-dependent host immunity by *S.* Typhimurium. On the other hand, the acquisition of zinc, iron, and magnesium from host cells is important for the intracellular growth of *S.* Typhimurium [20,22,35]. ZraP is a periplasmic molecular chaperone responsive to the presence of zinc, which has been shown to contribute to the zinc balance of *S.* Typhimurium [30]. So, the activation of *zraP* by YhjC might facilitate the uptake of zinc by intracellular *S.* Typhimurium. *S.* Typhimurium utilizes enterobactin and salmochelin to gain iron from the extracellular environment and IroC-dependent export of the siderophores, salmochelin and enterobactin, is necessary for the virulence of *S.* Typhimurium [19]. Moreover, IroD and IroE are the esterases that degrade siderophores to release iron within the bacteria [36,37]. Thus, the activation of *iroC*, *iroD*, and *iroE* by YhjC facilitates the uptake of iron by intracellular *S.* Typhimurium. MgtR is a highly hydrophobic peptide that can degrade membrane protein MgtC by binding to it, and, additionally, it can restrict the protein level of MgtA [38]. As MgtC and MgtA are responsible for the acquisition of magnesium by *S.* Typhimurium in macrophages [31,39], the repression of *mgtR* by YhjC increased expression of *mgtC* and *mgtA*, thus facilitating the uptake of magnesium by intracellular *S.* Typhimurium.

Four structural genes, *spvA*, *spvB*, *spvC* and *spvD*, form an operon, *spvABCD*. The operon *spvR* is a gene upstream of the operon *spvABCD*, and the gene transcription of *spvABCD* is positively regulated by SpvR. The transcription of the operon *spvABCD* begins at two tightly connected promoter sites, upstream of *spvA*, which are essential for the expression of four downstream structural genes [40,41]. Our transcriptome sequencing results, and previous studies, found that the mRNA abundance of *spvA*, *spvB*, *spvC*, and *spvD* decreased successively [42], suggesting that the transcription of *spvD* and other genes is regulated by other factors. In addition, there are long intergenic regions between the *spvA*, *spvB*, *spvC* and *spvD* genes on the chromosome of *S.* Typhimurium, such as the 260 base intergenomic region between *spvC* and *spvD*, which provide the basis for YhjC to directly bind the 5′ intergenic sequence of *spvD*. The LTTR regulatory protein is known to bind to the T–N_11_–A binding motif [43]. The 5′ intergenic region of *spvD* contains multiple T–N_11_–A binding motifs, which provide a basis for YhjC to bind to the 5′ intergenic region of *spvD*. Moreover, multiple T–N_11_–A binding motifs were also presented in the 5′ intergenic region of *zraP.* A study reported that the transcription of gene *iroB*, upstream of *iroCDE*, was negatively regulated by the global regulator Fur in response to iron depletion; however, the regulator promoting the transcription of *iroBCDE* has not yet been reported [44]. The regulatory mechanisms of *mgtR*, *iroCDE* by YhjC require further exploration in the future.

## 4. Materials and Methods

### 4.1. Bacterial Strains and Plasmids

*Salmonella* Typhimurium 14028S, obtained from American Type Culture Collection (ATCC), was used as the wild-type strain in this study. The mutant strain was constructed by using a λ Red homologous recombination system provided by a pSim17 plasmid encoding three proteins (Exo, Beta, and Gam), which are required for homologous recombination [45]. DNA fragments composed sequentially (5′–3′) of an upstream 50 bp sequence of the target gene, the gene sequence for chloramphenicol acetyltransferase expression, and the downstream 50 reverse complementary sequence of the target gene were amplified by PCR using Spark Pfu PCR Master Mix (Sparkjade, Jinan, China). The chloramphenicol-resistant pKD3 plasmid encoding chloramphenicol acetyltransferase was extracted by using SPARKeasy Superpure Mini Plasmid Kit (Sparkjade, Jinan, China) and used as the template for PCR, and primers were designed to cover the 50 bp homologous arm sequences. Next, the PCR products were electrophoresed on agarose gel and purified by SPARKeasy Gel DNA Extraction Kit (Sparkjade, Jinan, China). The DNA fragments were incorporated into competent bacterial cells by electroporation. After which, the bacterial cells were cultured in 1 mL 2YT medium (16 g/L tryptone, 10 g/L yeast extract, 5 g/L NaCl) for 2 h and spread on LB agar (10 g/L tryptone, 5 g/L yeast extract, 10 g/L NaCl, 20 g/L agar) containing 25 μg/mL chloramphenicol. Tryptone, yeast extract, and agar used in this study were all produced by Oxoid, Thermo Scientific™ (Waltham, MA, USA). The single colony grown on the LB agar was the potential target mutant, which was further identified using colony PCR and confirmed by Sanger sequencing.

For the construction of the complemented strain, primer pairs (Table 2) were designed to amplify DNA fragments consisting of the complete target gene sequence and its 300 bp upstream promoter sequence. Both the forward and reverse primer carries an enzyme restriction site and its protective base sequence. The PCR products were purified and re-circularized. The obtained DNA fragments and pWSK129 plasmid were both digested with *Bam*HI and *Eco*RI, simultaneously. The enzyme-digested products were purified and then ligated using T4 DNA ligase. The ligated product was introduced into the competent mutant strain by electrotransformation, and the bacterial cells were spread on LB agar containing 50 μg/mL kanamycin. The identification procedures were similar to that of the mutant strain construction.

For the construction of *Escherichia coli* BL21 (DE3) expressing YhjC protein with MBP tag, the whole gene sequence of *yhjC* was cloned into a pMAL-c5X plasmid between the *Nco*I and *Bam*HI sites. According to our experiment, YhjC protein had low solubility, and, therefore, the MBP tag was used to improve the solubility of the YhjC protein. The recombinant pMAL-c5X–yhjC plasmid was transformed into DE3, and the single colony growing on LB agar containing 100 μg/mL ampicillin was identified using the method described in mutant strain construction.

### 4.2. Determination of Bacterial Growth Curves

Different strains were inoculated in 20 mL LB medium (10 g/L tryptone, 5 g/L yeast extract, and 10 g/L NaCl) at a ratio of 1:1000 and cultured at 37 °C under shaking at 180 rpm for 12–16 h. Then the OD_600_ of different bacterial cultures was determined by a multi-mode microplate reader (TECAN, Männedorf, Switzerland). By adding fresh LB medium, the OD_600_ of different bacterial cultures was adjusted to 0.6. Then, the bacterial cultures were diluted 100 times in the new 100 mL LB medium or DMEM medium containing 10% FBS (Gibco, CA, USA), and the diluted bacterial cultures were mixed upside-down. The bacteria were cultured at 37 °C under shaking at 180 rpm. The OD_600_ of the different bacterial cultures was measured every 30 min, and the total culture time was 24 h.

### 4.3. Intracellular Replication Experiment

RAW 264.7 cells were cultured in the 12-well cell culture plate for 24 h before the experiment to obtain a cell monolayer. Different strains of S. Typhimurium were inoculated in 20 mL LB medium and cultured at 37 °C under shaking at 180 rpm for 12–15 h to reach the stationary phase. Then, the bacterial culture was sub-cultured in 20 mL fresh LB medium with an inoculation ratio of 1:100 for 12 h. The bacterial cultures were diluted by DMEM (containing 10% FBS), based on an MOI of 10 (bacteria number: macrophage number = 10). In order to reduce the surface charge of bacteria, the diluted bacterial cultures were shaken for 30 min at 37 °C at 180 rpm rotation speed. Then, 1 mL of the diluted bacterial cultures was gently added along the well wall. The cell culture plates were then centrifuged at 1000 rpm for 5 min, to allow the bacteria full contact with macrophages, and then put in an incubator containing 5% CO_2_ at 37 °C for 40 min (invasion and phagocytosis process). Next, the cell culture plate was quickly turned over, and the medium inside was drained away. Then, 1 mL phosphate buffered saline (PBS, Sparkjade, Jinan, China) was gently added to the well, and, subsequently, gently drained away to wash away the residual bacteria outside the macrophages. The washing operation was repeated one more time. To kill the extracellular bacteria, DMEM containing 100 µg/mL gentamicin was added to the cell culture well and placed in the incubator, at which point hours post-infection (hpi) was denoted as 0. After 1.5 h, the cells were washed twice with PBS, and then DMEM containing 20 µg/mL gentamicin was added to the cell culture well. At 2 or 16 hpi, the cell culture wells were washed 3 times with PBS, and then 1 mL 0.1% Triton X-100 solution was immediately added. The macrophages at the bottom of the cell culture plate were scraped off, and then sucked into tubes. The cell lysates were diluted in a series of gradients (10^3^, 10^4^, 10^5^) with PBS and spread on LB agar. Through overnight incubation at 37 °C, the CFU number of bacteria could be counted. The intracellular replication fold was calculated as the number of CFU at 20 h hpi divided by the number of CFU at 2 h [46].

### 4.4. Mouse Infection Experiment

Animal experiments in this study complied with the regulations of the Laboratory Animal Management and Use Committee of Nankai University. BALB/c mice (female, 7 weeks old, about 25 g) were purchased from Beijing Vital River Laboratory Animal Technology Co., Ltd. (Beijing, China). Mice were reared in ventilated cages with free access to food and water on racks. The mice were kept in a room with a temperature of about 25 °C and a natural light cycle.

To determine the CFU number of Salmonella colonized in the liver and spleen during the systemic infection of mice, and the survival rate of mice infected by Salmonella, the bacteria were cultured in 20 mL LB medium overnight and diluted to 1 × 10^5^ CFU/mL with normal saline (0.9% NaCl solution). A volume of 100 µL of the diluted bacterial suspension was injected into the right peritoneal cavity of the mice. For the calculation of mouse survival rate, the number of dead mice was recorded each day. With regard to the CFU count of Salmonella colonized in the liver and spleen, about 0.2 g of liver and entire spleen were harvested from the mice, and weighed, 3 days after intraperitoneal infection with Salmonella. Then, 1 mL of PBS was added to the tubes containing the tissues, which were then homogenized with a tissue homogenizer. The liquid homogenates were diluted by incrementally (10, 100, 1000) and spread on LB agar. The number of bacterial CFU was counted, and the CFU number per gram of organ was calculated [47].

### 4.5. RNA Extraction and Purification

Purified RNA was required for transcriptome sequencing and real-time quantitative PCR (RT–qPCR) analysis. Firstly, Salmonella was inoculated in 20 mL N-minimal medium (7.5 mM [NH_4_]_2_SO_4_, 1 mM KH_2_PO_4_, 5 mM KCl, 0.5 mM K_2_SO_4_, 10 μM MgCl_2_, 10 mM Tris-HCl (pH 7.5), 0.5% (v/v) glycerol, and 0.1% (w/v) casamino acids) and cultured at 37 °C under shaking at 180 rpm for 12–15 h. Then, the bacterial culture was re-inoculated in the new N-minimal medium with an inoculation ratio of 1:100. After 6 h, the bacterial culture was centrifuged at 10,000 rpm for 3 min, then the supernatant was poured away. The tubes containing bacterial pellets were immediately frozen in liquid nitrogen. In this experiment, the total bacterial RNA was extracted by using the traditional TRIzol reagent, according to the manufacturer’s instructions (Invitrogen, Waltham, MA, USA). Generally, bacteria collected from 40 mL N-minimal medium could meet the requirements of an RNA extraction. The RNeasy Mini Kit (QIAGEN, Germantown, MD, USA) was used to remove residual DNA from total RNA samples. The RNA concentrations were determined by using a NanoDrop 2000 spectrophotometer (Thermo Fisher Scientific, Waltham, MA, USA), and an OD_260_/OD_280_ and OD_260_/OD_230_ of the RNA solution more than 1.8 signified that the sample was not contaminated by proteins or carbohydrates, respectively [48].

### 4.6. Reverse Transcription and Real-Time Quantitative PCR

The extracted RNA was transcribed to cDNA using the PrimeScript RT Master Mix (Perfect Real Time) (TaKaRa, Shiga, Japan). The reverse transcription system (30 μL) was composed of 6 μL 5X PrimeScript RT Master Mix, x μL RNA solution (1.5 μg), and 24–x μL enzyme-free water. The liquids were flicked to mix well and centrifuged to the bottom of the microtubes; the reaction conditions were 37 °C for 15 min and 85 °C for 15 s.

The RT–qPCR primers were designed based on the target gene sequence (Table 2). The cDNA solution was pre-mixed with nuclease-free water, primer solutions were added to PowerUp™ SYBR™ Green Master Mix (Applied Biosystems, Waltham, MA, USA), and then both mixtures were then added to the 96-well reaction plate. The 30 μL RT-qPCR reaction system consisted of 15 μL PowerUp™ SYBR™ Green Master Mix, 1 μL cDNA (approximately 50 ng), 1 μL forward primer and reverse primer (initial concentration of 10 μM each), and 12 μL nuclease-free water. The reaction plate was sealed with sealing film, and, finally, the reaction plate and its holder were centrifuged for a short time, so that all the liquid flowed into the bottom of the wells. The 96-well reaction plate was put into the QuantStudio™ 5 real-time fluorescence quantitative PCR instrument (Applied Biosystems, Waltham, MA, USA) [49]. The program used for RT–qPCR is shown in Table 3.

### 4.7. Library Construction and Transcriptome Sequencing Analysis

Before the library was constructed, the extracted RNA was subjected to agarose gel electrophoresis to verify the degradation of the RNA. Generally, good quality RNA has at least two clear bands of ribosomal RNA (rRNA). If two distinct bands of 16S and 23S rRNA were found in the bacterial RNA extract after electrophoresis, the RNA integrity was good and hardly degraded.

The NovaSeq 6000 System of Illumina sequencing platform (Illumina, San Diego, CA, USA) was used for RNA sequencing, and the NEBNext Ultra Directional RNA Library Prep Kit (NEB, USA) was used to construct the cDNA library. For specific experimental operations, we referred to the instructions of the kit. The read numbers obtained by sequencing were compared with the reference genomes of the wild-type species. Analysis of the expression level of a gene was performed by HTSeq software (version 0.6.1). FPKM values were calculated using HTSeq software to represent the gene expression level, where FPKM represented fragments per kilobase of transcript per million fragments mapped. DESeq2 software (v1.6.3) was used to compare the difference in gene expression between the two groups of samples. Fold changes in gene expression of ≥2 and *p* < 0.05 were regarded as indicating a differentially expressed gene (DEG) [33,50].

### 4.8. Immunofluorescence Microscopy

To visualize S. Typhimurium in host cells, RAW 264.7 cells were cultured on a round coverslip in cell culture plates and infected as per the infection procedure described in this study. At the specified infection time, DMEM was aspirated and discarded, and the cells were gently rinsed with cooled PBS 3 times. Then, the cells were fixed with 4% paraformaldehyde for 10 min and rinsed with cooled PBS 3 times (3 min each time). Next, the cells were permeated with 0.5% Triton X-100 for 20 min, then rinsed with PBS 3 times, blocked with 5% BSA (dissolved in PBS) for 30 min, and rinsed with PBS again. Subsequently, the cells were incubated with mouse anti-Salmonella antibody labeled with FITC (ab20320, Abcam, 1:80 dilution) for 1 h and rinsed with PBS 3 times. Finally, the cells were incubated with DAPI (D21490, Invitrogen) for 2 min. After gentle rinsing twice, the sealer was added to the slide. The round coverslips were peeled from the cell culture plates with clean tweezers and placed upside down on the glass slides. Cell images were obtained using a laser confocal scanning microscope (LSM800, Zeiss) and analyzed with ZEN 2.3 (blue version) [50].

### 4.9. Purification of Recombinant Proteins

BL21 bacteria harboring recombinant the pMAL–c5X–yhjC plasmid were inoculated in 20 mL LB medium containing 100 μg/mL ampicillin and cultured overnight at 37 °C under shaking at 180 rpm. The bacteria were sub-cultured in 500 mL fresh LB medium with an inoculation ratio of 1:100 for 3 h. When the OD_600_ of the bacteria was around 0.6, 1 mM final concentration of isopropyl β-d-thiogalactoside (IPTG) was added to induce the expression of MBP-tag protein. After 6 h of culture, the bacteria culture was placed in a crushed ice to cool for 15 min, and then centrifuged at 5500 rpm at 4 °C for 5 min. The supernatant was poured away, and then the bacteria was washed with PBS. After the collection of bacterial pellets, the protein was immediately purified [51].

In this study, amylose resin (NEB, USA) was used to purify the MBP-tag protein. Amylose resin is composed of amylose and agarose beads, which have good binding ability for the MBP label protein. Firstly, the binding buffer and elution buffer were prepared. The binding buffer was composted of 20 mM Tris-HCl (pH 7.4), 200 mM NaCl, and 1 mM dithiothreitol (DTT), and the elution buffer was the binding buffer supplemented with 10 mM maltose. The gravity column was fixed vertically, and 2 mL amylose resin was added to the column. When the protective liquid was drained from the amylose resin, 15 mL binding buffer was added into the column to balance the amylose resin. Next, 1–5 mL binding buffer was added to every 100 mg bacteria (wet weight), and then PMSF (final concentration 1 mM), DNase I (final concentration 1 U/mL), lysozyme (final concentration 0.1 mg/mL), and RNase A (final concentration 10 μg/mL) were added. Bacterial pellets were homogenized in the binding buffer and lysed with ultrasonic treatment. After centrifugation at 10,000 rpm for 10 min, the supernatant of the bacterial lysate was mixed with the same volume of binding buffer and added into the column containing amylose resin in a 4 °C environment. In order to reduce the non-specific adsorption of proteins, 20 mL binding buffer was added to the column. Then, 20 mL elution buffer was added into the column and the outflow was collected until the OD_280_ was stable. Target proteins were detected and identified by SDS-PAGE and Coomassie blue staining. The column containing amylose resin could be saved after the washing procedures and could be regenerated by washing with 0.1% SDS or 0.5 M NaOH solution.

### 4.10. Electrophoretic Mobility Shift Assay

PCR primers were designed to amplify the promoter DNA fragments composed of 300 bp upstream and the first 100~200 bp of the target gene (Table 2). Moreover, a pair of primers were designed to amplify 16S rDNA fragments (as negative control). The PCR products were electrophoresed on agarose gel and recovered. The EMSA binding buffer used in this study was made up of 10 mM Tris-HCl (pH 7.5), 100 mM NaCl, 1 mM DTT, and 10% glycerol (prepared with nuclease-free water).

The 6% native gel was prepared and its formula was 6% acrylamide (37.5:1), 0.067% ammonium persulfate (AP), and 0.1% N,N,N′,N′-Tetramethylethylenediamine (TEMED, Sparkjade, Jinan, China) in 0.5 × Tris-borate-EDTA (TBE, Sparkjade, Jinan, China) buffer. Each tube received 100 ng DNA followed by increasing volumes of purified protein to reach different final protein concentrations. Then, different volumes of EMSA binding buffer were added to make the total volume up to 40 μL. The mixtures were slowly aspirated with a pipette and incubated at room temperature for 20 to 30 min. Then, 0.5× TBE buffer was added to the electrophoresis tank as the electrophoresis buffer. Then, 20 μL of sample was added into each well of the 6% polyacrylamide gel. The electrophoresis was performed at 90 V for 2 h. After electrophoresis, the glass plate holding the gel was carefully opened and dipped into deionized water. Through moving up and down, the gel was released from the glass plate. Gels were stained with GelRed solution for 10 min, and an automatic gel imager (Tanon, Shanghai, China) was used to generate images showing the DNA bands [45].

### 4.11. Statistical Analysis and Plotting

GraphPad Prism (version 8.0.1) was used for statistical analysis and plotting. Unless otherwise noted, data shown in the figure or table were from three independent biological experiments (in vitro assays) or a combination of two separate biological experiments (in vivo assays). According to the requirements of statistical analysis, Student’s t-test, log-rank (Mantel–Cox) test, or Mann–Whitney U test, was used to analyze the significant differences between the two groups. A value of p < 0.05 meant significant difference (* *p* < 0.05; ** *p* < 0.01; *** *p* < 0.001; **** *p* < 0.001).

## 5. Conclusions

The putative regulator of S. Typhimurium, YhjC, promotes the transcription of spvD and zraP genes by directly binding to the promoter sequences of these two genes; YhjC also regulates the transcription of iroCDE and mgtR indirectly. It has been reported that spvD is related to inhibition of the NF–κB signaling pathway, and that zraP, iroCDE, and mgtR are involved in the homeostasis of zinc, iron, and magnesium, respectively. Therefore, YhjC facilitates the replication of S. Typhimurium in macrophages and the systemic infection of mice by regulating the transcription of multiple virulence-related genes. YhjC could represent a promising target to control S. Typhimurium infection. Limitations of the study included the following: the precise binding site of YhjC in spvD and zraP promoters were not indicated; and the regulatory mechanisms of iroCDE and mgtR by YhjC were not indicated. More experiments are required to fully understand the regulatory mechanisms of YhjC in S. Typhimurium.

## Figures and Tables

**Figure 1 ijms-24-01302-f001:**
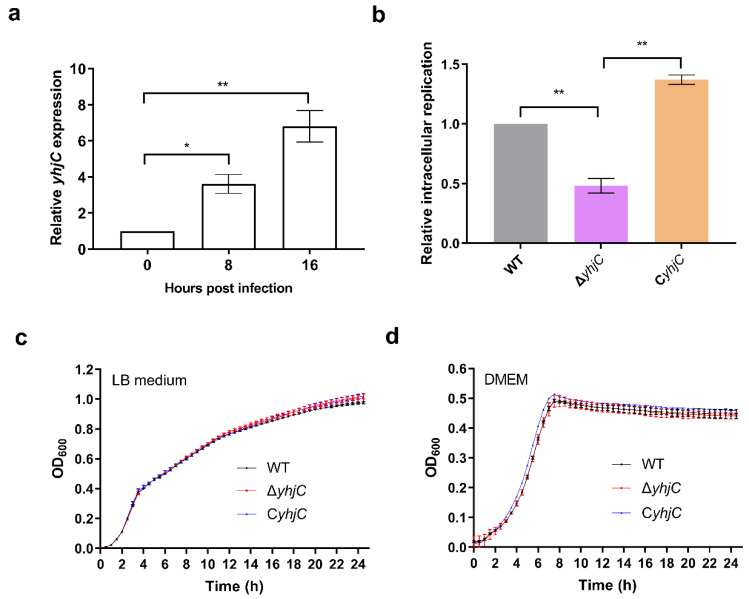
Deletion of yhjC reduced the replication ability of S. Typhimurium in RAW 264.7 cells. (**a**) Relative expression of yhjC gene of S. Typhimurium before and after entering 264.7 cells, where 0 represents the gene expression of S. Typhimurium cultured in DMEM medium for 8 h. (**b**) Fold replication of S. Typhimurium WT, ΔyhjC (yhjC mutant), and CyhjC (complemented strain) in RAW264.7 cells at 16 h post-infection. (**c**,**d**) Growth curves for S. Typhimurium WT, ΔyhjC, and CyhjC cultured in LB medium (**c**) and DMEM (**d**). The OD_600_ of the bacterial suspensions were measured every half hour using a multifunctional microplate reader. (**a**–**d**) Data were generated from three independent experiments and are presented as mean ± standard deviation. The *p*-values were determined using unpaired Student’s *t*-test (* *p* < 0.05; ** *p* < 0.01).

**Figure 2 ijms-24-01302-f002:**
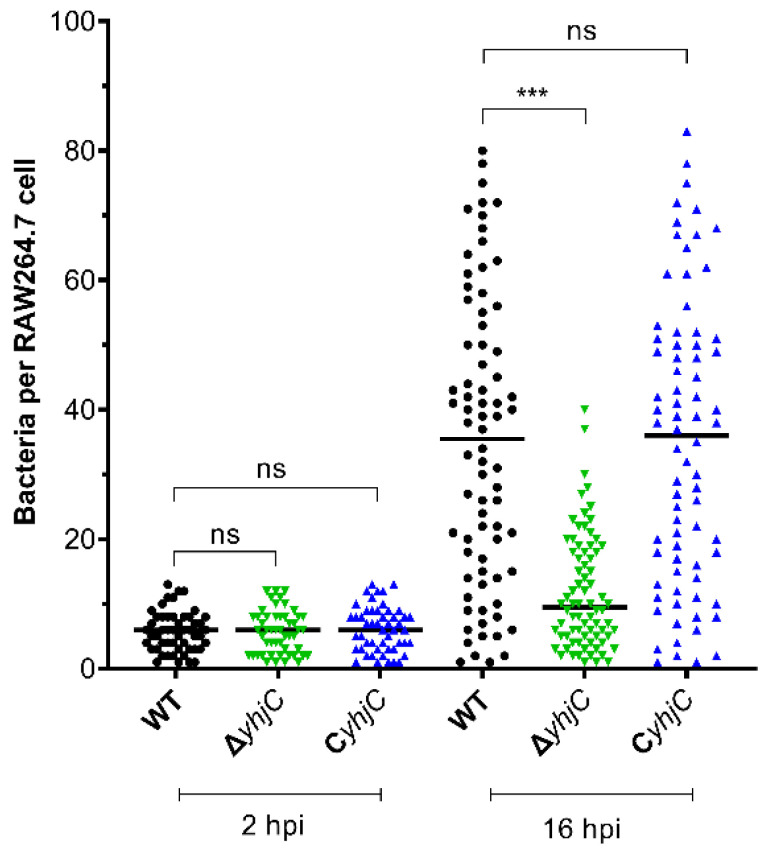
Numbers of S. Typhimurium cells in single RAW264.7 cell. At least 50 RAW264.7 cells in random fields were used to count the numbers of S. Typhimurium at 2 and 16 hpi. The *p* values were calculated by unpaired Student′s t test. *** *p* < 0. 001; ns, not significant.

**Figure 3 ijms-24-01302-f003:**
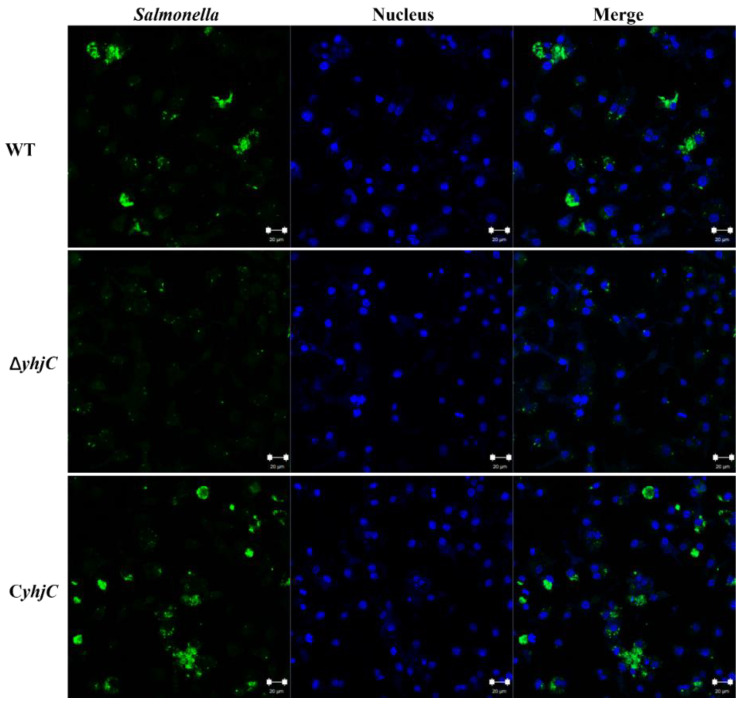
Representative images of RAW264.7 cells infected by WT, ΔyhjC, and CyhjC bacteria at 16 hpi. Bacteria were labeled with anti-Salmonella antibody (green), and cell nuclei were stained with 4′,6-diamidino-2-phenylindole (DAPI, blue).

**Figure 4 ijms-24-01302-f004:**
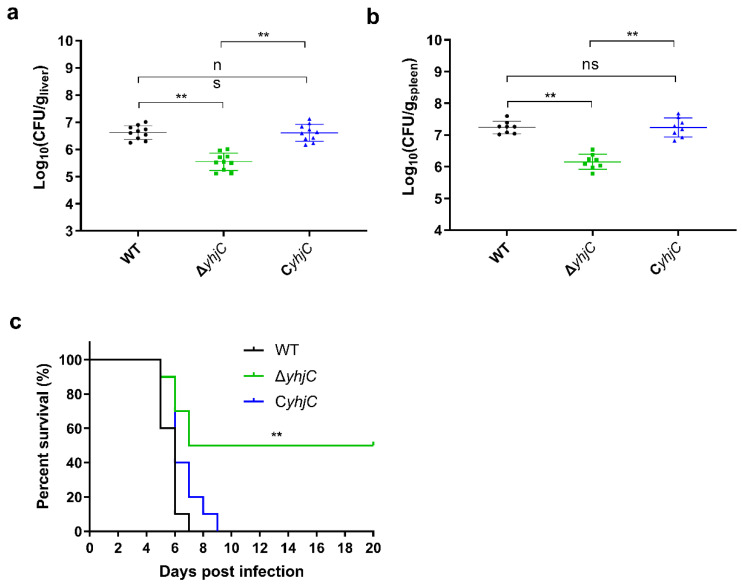
Deletion of yhjC reduced the virulence of S. Typhimurium to mice. (**a**,**b**) CFU numbers of WT, ΔyhjC and CyhjC in mouse liver (**a**) and spleen (**b**) at 3 days after intraperitoneal injection of S. Typhimurium. n = 10 mice/group. (**c**) The survival curves for mice infected with WT, ΔyhjC and CyhjC. n = 10 mice/group. (**a**,**b**) Data were obtained from three independent experiments and are presented as mean ± standard deviation. The Mann–Whitney U test (**a**,**b**) or log-rank (Mantel–Cox) test (**c**) was used for the calculation of p values, (** p < 0.01).

**Figure 5 ijms-24-01302-f005:**
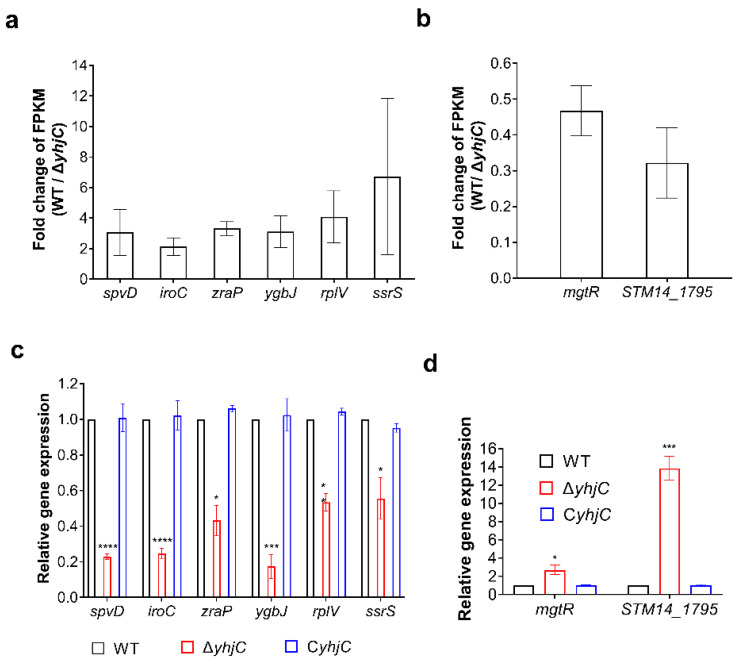
Validation of transcriptome sequencing data. (**a**,**b**) Fold change in the FPKM of representative differentially expressed genes (DEGs) (WT/Δ*yhjC*), taken from RNA sequencing data. (**c**,**d**) RT-qPCR was used to verify the gene expression differences among WT, Δ*yhjC* and C*yhjC*. Unpaired Student’s *t*-test was used to calculate *p* values (* *p* < 0.05; *** *p* < 0.001; **** *p* < 0.0001).

**Figure 6 ijms-24-01302-f006:**
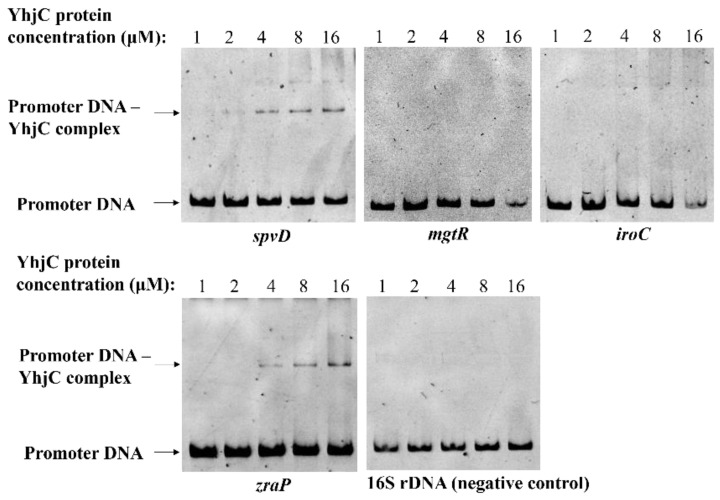
EMSA of purified YhjC protein with the 5′ intergenic region DNAs of target genes.

**Table 1 ijms-24-01302-t001:** Differentially expressed genes (DEGs) between WT and ΔyhjC.

Gene Name	Log_2_FC	Product
*ProP*	−3.532	MFS transporter
*STM14_RS03435*	−2.113	YkgB family protein
*ssrS*	−2.085	6S RNA
*zraP*	−1.739	zinc resistance sensor/chaperone ZraP
*rplV*	−1.694	50S ribosomal protein L22
*STM14_RS15610*	−1.618	SDR family oxidoreductase
*ygbJ*	−1.593	NAD(P)-dependent oxidoreductase
*ygbM*	−1.542	HPr family phosphocarrier protein
*spvD*	−1.438	SPI-2 type III secretion system effector cysteine hydrolase SpvD
*lhgO*	−1.277	L-2-hydroxyglutarate oxidase
*STM14_RS14715*	−1.242	hypothetical protein
*iroD*	−1.221	esterase family protein
*iroE*	−1.220	alpha/beta hydrolase
*ygbL*	−1.208	aldolase
*hypC*	−1.205	hydrogenase 3 maturation protein HypC
*csiD*	−1.187	carbon starvation induced protein CsiD
*Sbp*	−1.173	sulfate ABC transporter substrate-binding protein
*nrdF*	−1.126	class 1b ribonucleoside-diphosphate reductase subunit beta
*ygbK*	−1.096	3-oxo-tetronate kinase
*nadB*	−1.074	L-aspartate oxidase
*iroC*	−1.063	ABC transporter ATP-binding protein
*rplC*	−1.031	50S ribosomal protein L3
*STM14_RS16725*	−1.016	polysaccharide deacetylase
*rplD*	−1.008	50S ribosomal protein L4
*ompW*	1.038	outer membrane protein OmpW
*lysA*	1.111	diaminopimelate decarboxylase
*ygiW*	1.115	YgiW/YdeI family stress tolerance OB fold protein
*mgtR*	1.131	protein MgtR
*fljB*	1.137	FliC/FljB family flagellin
*STM14_RS06850*	1.165	DUF2441 domain-containing protein
*fdoI*	1.202	formate dehydrogenase cytochrome b556 subunit
*mgtA*	1.207	magnesium-translocating P-type ATPase
*narG*	1.252	nitrate reductase subunit alpha
*grcA*	1.277	autonomous glycyl radical cofactor GrcA
*fdxH*	1.312	formate dehydrogenase subunit beta
*fliI*	1.354	flagellum-specific ATP synthase FliI
*STM14_1795*	1.705	acid shock protein
*STM14_RS12610*	1.829	MFS transporter
*adiY*	2.469	helix-turn-helix domain-containing protein

Log_2_FC is based on the FPKM value of a gene in Δ*yhjC* versus that of WT.

**Table 2 ijms-24-01302-t002:** Primers involved in this study.

Targets		Primer Sequences (5′–3′)
		Primers used for the establishment of mutant strains
Δ*yhjC*	F	AGAAAATCAATAAAACCAGCCGCTTCAGCGTCACTGTTTCAATAAGAACAATAATAGAGCCTGATG GTGTAGGCTGGAGCTGCTTC
	R	GGTAGTCCTTCATCATTAATAGAATATGTCAGTATAGCTTCTCCCCGGTGGATGCTGAAAATGCGG CATATGAATATCCTCCTTAGTTC
		Primers used for RT-qPCR analysis
16S rRNA	F	ACTGGCAGGCTTGAGTCTTGTAGA
	R	GGCACAACCTCCAAGTAGACATCG
*spvD*	F	GAGAGTTTCTGGTAGTGCGTCATCCC
	R	CGCTGTCTAATCCCACTGTAGGAGAG
*zraP*	F	GTGGCAACAGGGAGGTAGCCC
	R	ACTGGCGGTCAGTAGCGCGT
*ssrS*	F	CCGCAGGCTGTAACCCTTGAAC
	R	GAGATGTTTGCAAGCGGGCC
*ygbJ*	F	CGGCGTCGTGGATGCGTTA
	R	CAGCACAGGCTTCAGGCGGG
*iroC*	F	GGTCAATAACGGCGGCACGG
	R	CGTCACCCTGGTCAGCGAGG
*rplV*	F	TCGCCTTGTTGCTGACCTGATTC
	R	CGCTTCATGCTCGGGCCTT
*mgtR*	F	CGCTCACCCGATAAAATCATCGC
	R	CGATTTGCCAGAGGGCTAAACAC
*STM14_1795*	F	GTTGTTGCCGCTGCAATGGG
	R	TGGTGCTGGGTGGTCTGGGT
		Primers for the construction of recombinant plasmids
pMAL-c5x-*yhjC*	F	CATGCCATGGGC ATGGATAAAATATATGCAATGAAATTGT
	R	CGGGATCC CTACTCTGCGGCCTCTTTAATG
pWSK129-*yhjC*	F	CGGGATCC AAATAGCCAATCCGCCTGAAA
	R	GGAATTC AAATAGCCAATCCGCCTGAAA
pWSK129-*spvD*	F	CGGGATCC TTTACGTGAGGAACCGTTTTATCG
	R	CCGCTCGAG TCAATCGTGTTTTTCATCATAAGCC
		Primers used for the amplification of DNA sequences used in EMSA experiments
*spvD*	F	TTTACGTGAGGAACCGTTTTATCG
	R	CTTTAATTCTCTTGACTTCATTTGAATCA
*mgtR*	F	CTGGTTTATTGAGGGTCTGCTCT
	R	TTAGAAAACGATTTGCCAGAGG
*zraP*	F	CTCAACCACGTTGCAGCGG
	R	CATTGTTGCCCCAGTGATGC
*iroC*	F	TGGCCAGGGTGCCGAC
	R	CTACGATGACAATGATGCTCAGTG
16S rDNA	F	AAATTGAAGAGTTTGATCATGGCTC
	R	GCATGGCTGCATCAGGCTT

**Table 3 ijms-24-01302-t003:** The program of real-time quantitative PCR.

Reaction Stages	Temperatures (°C)	Duration	Cycle Number
Activation of Uracil-DNA Glycosylase	50	2 min	No
Initial denaturation	95	2 min	No
Denaturation	95	15 s	40
Annealing/extension	60	1 min	

## Data Availability

All data are presented within manuscript.

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
