# Peer review of "The LysR-Type Transcription Regulator YhjC Promotes the Systemic Infection of Salmonella Typhimurium in Mice"

_ijms, 2023, doi:10.3390/ijms24021302_

Round 1

Reviewer 1 Report

The manuscript of of W. Li et al. titled The LysR-Type Transcription Regulator YhjC Promotes the Systemic Infection of Salmonella Typhimurium in Mice is devoted to the role of LysR-Type Transcription Regulator YhjC in S. Typhimurium infection in mice. In general, the manuscript is well written I didn't find major issues with it.

The minor issues and some suggestions are the next ones:

(Since the lines are not numbered I give the page number and line in the section)

Pg 1, line 3 in Introduction - I suggest to change as Salmonella enterica subsp. Enterica serovar Typhimurium

Pg 2 - Add what is ROS and RNS

Pg 8 - Add what is EMSA

Pg 10 - Add from where Salmonella Typhimurium 14028S was obtained

Pg 10 - Replace "recycled" with "recircularized"

Pg 11 - Add "and" between NcoI BamHI

Pg 12: Add manufacturers for LB and DMEM media

Pg 12: Add name/model of spectrophotometer

Pg 12: Add which volumes was used for bacteria culturing

Pg 13: Add name and model of "PCR instrument"

Pg 13: Omit "Finally, the PCR program was set, and the reaction was started." Replace it with program used for RT-qPCR

Pg 13: Measurement of RNA concentration (5.5) and RNA quality (5.7) by NanoDrop 2000 were the same thing or different?

Pg 14, Line 1: Add model of Illumina sequencer used

Pg 15: Add name and model of "automatic gel imager"

Pg 15: Italicize in vitro and in vivo

Reviewer 2 Report

The manuscript by Li et al. is well designed and conducted, perfectly follow the journal policy, and add valuable data on the knowledge of YhjC protein contribution, as putative regulator, in the S. Typhymurium virulence via the regulation of multiple virulence genes. I support its possible further processing after appropriate minor modifications as mentioned below:

Abstract

„expression of yhjC” – unclear, please indicate what you mean yhjC, it is a gene? (similar request for YhjC – it is a protein?)

„We observed that...” – please avoid the using of personal mode formulations, it is not so characteristic for the scientific style. Please revise it throughout the manuscript

At the end of the abstract section, please indicate the practical application of the main findings.

Introduction

„[1, 2].” – please delete the space delimitation within the numbers when you indicate references in brackets, throughout the manuscript

„... leading to the death of the host.” – please indicate an adequate reference at the end of the paragraph

Results

“CFU” – being the first appearance in the text, please indicate the meaning of this acronym

Discussion

The overall impression of the reader is the fact that the discussion of the obtained valuable results in terms of comparison with previously reported findings needs improvements. In this regards, please try to improve this section consulting and citing at least five new references.

Conclusions

The authors must highlight the study limitations and further perspectives in this research area

Materials and methods

„wild-type strain” – please provide the origin of the used S. Typhimurium strain

„2YT medium” „LB agar” – I wonder, the used reagents were prepared „in house” or were commercially procured? If the second option is true, and in order to increase the study reproducibility, please uniformly provide for each of the used reagents the production company name, city and country throughout the manuscript!!!

I have not identified any bibliographic reference within the materials and methods section! I wonder, why?
